# StepHint: Multi-level Stepwise Hints Enhance Reinforcement Learning to Reason

## Abstract

Reinforcement learning with verifiable rewards (RLVR) is a promising approach for improving the complex reasoning abilities of large language models (LLMs). However, current RLVR methods face two significant challenges: the near-miss reward problem, where a small mistake can invalidate an otherwise correct reasoning process, greatly hindering training efficiency; and exploration stagnation, where models tend to focus on solutions within their "comfort zone," lacking the motivation to explore potentially more effective alternatives. To address these challenges, we propose StepHint, a novel RLVR algorithm that utilizes multi-level stepwise hints to help models explore the solution space more effectively. StepHint generates valid reasoning chains from stronger models and partitions these chains into reasoning steps using our proposed adaptive partitioning method. The initial few steps are used as hints, and simultaneously, multiple-level hints (each comprising a different number of steps) are provided to the model. This approach directs the model's exploration toward a promising solution subspace while preserving its flexibility for independent exploration. By providing hints, StepHint mitigates the near-miss reward problem, thereby improving training efficiency. Additionally, the external reasoning pathways help the model develop better reasoning abilities, enabling it to move beyond its "comfort zone" and mitigate exploration stagnation. StepHint outperforms competitive RLVR enhancement methods across six mathematical benchmarks, while also demonstrating superior generalization and excelling over baselines on out-of-domain benchmarks.

## 1 Introduction

Eliciting the reasoning capabilities of large language models (LLMs) through Reinforcement Learning with Verifiable Rewards (RLVR) has emerged as a powerful paradigm (Jaech et al., 2024; Guo et al., 2025). In RLVR frameworks, a policy model explores the solution space by generating reasoning chains. The model is then optimized using algorithms like PPO (Schulman et al., 2017) and GRPO (Shao et al., 2024), based on the advantages of final outcomes of these chains.

However, free exploration within the vast and complex solution space introduces significant challenges. A key issue is the **near-miss reward problem**, where a single incorrect step voids an otherwise reward-worthy reasoning chain. This leads to training inefficiency, as models expend resources on repeatedly almost-correct solutions. Moreover, as shown by Yue et al. (2025), existing RLVR methods often refine the model's ability to sample known reasoning chains rather than discover novel or higher-quality ones. Consequently, when a task exceeds the model's current capabilities, it tends to remain confined to its "comfort zone," unable to independently advance beyond familiar solutions—an issue we term **exploration stagnation**.

We propose StepHint, a novel augmented RLVR algorithm that integrates multi-level stepwise hints to address these challenges. StepHint leverages reasoning chains from advanced models such as Deepseek-R1 (Guo et al., 2025), partitioning them into discrete reasoning steps.[1] It then provides only the initial few steps as hints for the model to complete the reasoning process. This approach effectively simplifies the solution space while preserving sufficient exploratory flexibility. Specif-

---

[1] A *reasoning step* refers to a distinct logical stage within the overall reasoning chain and typically comprises multiple tokens. It should not be confused with a *token-prediction step* during generation or a *training step*.

ically, during RLVR—regardless of the policy optimization algorithm used—StepHint's pipeline comprises two key steps:

**Step 1: Adaptive stepwise partitioning of reasoning chains.** We introduce a probabilistic partitioning strategy that adaptively splits reasoning chains into meaningful steps, moving beyond superficial markers like "First" or "Second." Our method estimates, at each token, the model's probability of generating an end-of-reasoning token (e.g., `</think>`). A position is identified as a candidate endpoint if the estimated probability at this position exceeds that of the next position. From these candidates, we randomly sample several endpoints, subject to a minimum length constraint, resulting in a fixed number of segments. The initial few steps are later provided as hints to guide the model's rollouts during RL training.

**Step 2: Multi-level hints for problem solving.** We define a hint's "level" as the number of initial reasoning steps it provides. A high-level hint contains many steps, potentially making the task trivial for the model, diminishing training efficacy. Conversely, a low-level hint with too few steps may be insufficient to guide the model, leaving it vulnerable to the "near-miss reward problem." Determining the optimal hint level for a given model-problem pair is inherently difficult, as the model abilities keeps improving durning trianing. Our simple yet effective solution is to generate multi-level hints for each problem. With sufficiently fine-grained step partitioning, at least one hint level is likely to be suitable for the model's current reasoning ability.

By adaptively providing multi-level hints, StepHint effectively addresses both the near-miss reward problem and exploration stagnation. First, the model receives appropriate guidance to complete reasoning chains correctly, significantly reducing near-miss rewards and improving training efficiency—leading to faster convergence. Second, exposure to high-quality hints steers the policy toward more sophisticated reasoning patterns, preventing stagnation during independent exploration. This not only enhances the model's ability to break through its "comfort zone" but also avoids the poor generalization typical of SFT-based methods.

We evaluate StepHint by training a series of LLMs on mathematical tasks and comparing their performance against strong RLVR-enhanced baselines. Results demonstrate StepHint's effectiveness on both in-domain (math) and out-of-domain tasks.

• **In-domain performance**: Across six math benchmarks, StepHint surpasses existing methods by an average accuracy of 3.16 percentage points. Notably, it achieves significant improvements in pass@$k$ performance—a rigorous measure of reasoning abilities (Yue et al., 2025)—on two challenging benchmarks, AIME24 and AIME25 (Li et al., 2024), even at large $k$ values.

• **Out-of-domain generalization**: StepHint also achieves the highest results on out-of-domain, non-mathematical benchmarks such as ARC-C (Clark et al., 2018) and GPQA-D (Rein et al., 2024), highlighting its robust generalization beyond its training domain.

## 2 BACKGROUND: REINFORCEMENT LEARNING WITH VERIFIABLE REWARDS

Reinforcement Learning (RL) has been instrumental in advancing the reasoning capabilities of Large Language Models (LLMs) by enabling them to learn optimal reasoning chains through reward-based feedback (Hu et al., 2025; Guo et al., 2025). A popular paradigm in this domain is Reinforcement Learning with Verifiable Rewards (RLVR), which is a specialized RL paradigm for training LLMs on tasks where the correctness of an outcome can be objectively verified, such as mathematical problem-solving or code generation. In the RLVR framework, the learning process is typically driven by automated, often binary (correct/incorrect), reward signals, which facilitates scalable self-improvement (Gao et al., 2023).

**Proximal Policy Optimization (PPO)** PPO (Schulman et al., 2017) is a widely-used algorithm that optimizes the the LLM's generation policy ($\pi_\theta$) by maximizing expected rewards while preventing excessively large updates that could destabilize training. Given a problem $q$, the policy model $\pi_\theta$ samples $N$ rollouts, denoted as $\{y_1, y_2, \cdots, y_N\}$, PPO optimizes the following objective:

$$\mathcal{L}_{\theta}^{PPO} = \frac{1}{N} \sum_{i=1}^{N} \frac{1}{|y_i|} \sum_{t=1}^{|y_i|} \left\{ \min \left[ r_{i,t} \hat{A}_{i,t}, \text{clip} \left( r_{i,t}, 1 - \epsilon, 1 + \epsilon \right) \hat{A}_{i,t} \right] \right\} - \beta D_{KL}(\pi_{\theta} || \pi_{\text{ref}}),$$

where:

- $r_{i,t} = \frac{\pi_{\theta}(y_{i,t}|q, y_{i,<t})}{\pi_{\text{old}}(y_{i,t}|q, y_{i,<t})}$ is the probability ratio between the current policy $\pi_{\theta}$ and the policy before the update, $\pi_{old}$. The clip function bounds this ratio, which disincentivizes overly aggressive policy changes that could destabilize training.

- $\hat{A}_{i,t}$ is the advantage of taking token $y_{i,t}$ as the $t$ token of rollout $i$. It quantifies how much better that action is compared to the average action value at that state.

- $\beta D_{KL}(\pi_{\theta} || \pi_{\text{ref}})$ is a penalty term that discourages the current policy from deviating too far from a reference policy $\pi_{\text{ref}}$ (often the initial model).

A key component of standard PPO is the critic model, which is required to calculate the token-level advantage $\hat{A}_{i,t}$. However, training this critic is computationally expensive and complex.

**Group Relative Policy Optimization (GRPO)**    To address the challenges of PPO, GRPO (Shao et al., 2024) is introduced as a simpler yet effective alternative. It has demonstrated strong performance and efficiency, particularly in complex mathematical reasoning tasks (Liu et al., 2025; Yan et al., 2025; Zeng et al., 2025; Hu et al., 2025).

GRPO computes a single, uniform advantage value for all tokens within a rollout, based on the final outcome of that entire rollout. Specifically, for a given problem $q$, $N$ rollouts $\{y_1, y_2, \cdots, y_N\}$ are sampled. Each complete rollout $y_i$ is assigned a final reward $R(y_i)$, which is typically binary in RLVR settings: $R(y_i) = 1$ if the answer of $y_i$ is correct, and $R(y_i) = 0$ otherwise. GRPO then calculates a rollout-level advantage by normalizing this reward across the group. This advantage value is assigned to every token within that rollout:

$$\hat{A}_{i,t}^{GRPO} = \frac{R(y_i) - \text{mean}\left(\{R(y_1), \cdots, R(y_N)\}\right)}{\text{std}\left(\{R(y_1), \cdots, R(y_N)\}\right)}. \tag{1}$$

By replacing the token-level advantage $\hat{A}_{i,t}$ with this rollout-level advantage $\hat{A}_{i,t}^{GRPO}$, GRPO retains the core PPO objective while eliminating the need for a critic model.

Since PPO and GRPO have become the predominant methods for training LLMs on reasoning tasks, this paper will focus exclusively on these two approaches.

## 3 METHOD

We frame the reasoning process as a stepwise reduction of a solution space. This perspective inspires our core idea: guiding the model's exploration with stepwise hints. To build this foundation, we first formalize the generation of reasoning chains as a solution space exploration (Section 3.1). This formalization helps identify critical issues in existing methods, such as near-miss rewards and exploration stagnation, and lays the groundwork for our proposed method, StepHint (Section 3.2), which is designed to address these challenges.

### 3.1 MOTIVATION: A SOLUTION SPACE REDUCTION VIEW OF REASONING

We model the autoregressive generation of a reasoning chain as a sequential reduction of a solution space $\mathcal{R}$, which comprises all possible reasoning chains for a given prompt $\mathcal{C}$ (Guo et al., 2025). This process is represented by a sequence of states, where each state $S_k = (\mathcal{C}, t_1, \ldots, t_k)$ corresponds to the partial reasoning chain after $k$ tokens. The complexity or uncertainty of the remaining solution space at each state is quantified by the conditional entropy $H(\mathcal{R}|S_k)$.

While this entropy is generally intractable to compute directly, as it requires summing over all possible chains in $\mathcal{R}$, it serves as a powerful conceptual tool for analysis. A higher entropy indicates a complex and unconstrained solution space. The following proposition formalizes the intuition that each generation step, in expectation, reduces the solution space's complexity.

**Proposition 1.** *Let $\mathcal{R}$ be the solution space and $S_{k-1}$ be the state after $k-1$ tokens have been generated. Upon generating the next token $t_k$ to form state $S_k = (S_{k-1}, t_k)$, the expected entropy of the solution space is bounded by the current entropy:*

$$\mathbb{E}_{t_k \sim P(\cdot|S_{k-1})}[H(\mathcal{R}|S_k)] \leq H(\mathcal{R}|S_{k-1}).$$

We leave the detailed proof in Appendix B. Proposition 1 formally establishes that autoregressive generation is a structured process of uncertainty reduction. However, the reduction in entropy quantifies the convergence of certainty, not necessarily the correctness or logical validity of the reasoning chain. A model can become increasingly certain about a flawed conclusion, which still manifests as a decrease in entropy.

This distinction reveals a critical failure mode. An early error can irrevocably prune the subspace of correct solutions, $R^* \subset R$. Formally, this occurs when the model reaches a state $S_k$ where the probability of the correct solution set collapses to zero: $P(R^*|S_k) \approx 0$. Subsequently, the model may continue to confidently reduce entropy, but it does so within the incorrect subspace $R \backslash R^*$, inevitably arriving at a "confident but wrong" conclusion.

### 3.2 STEPHINT: MULTI-LEVEL STEPWISE HINTS ENHANCE RLVR

Based on the views above, an initial error can lead to an incorrect final answer, even if the subsequent reasoning is logically sound, as it originates from a flawed premise. This "near-miss" reward problem can be mitigated by providing early guidance. Furthermore, a model's exploration of the solution space is intrinsically constrained by the model's ability. Without external guidance, the exploration will be limited to a narrow subspace (Yue et al., 2025). Our proposed method, StepHint, addresses these two challenges by leveraging part of the strong reasoning chains from a more capable model as hints during training.

Given a problem, StepHint first obtains a valid reasoning chain from a stronger model, and then performs two key stages to enhance the target model being trained: (1) adaptive stepwise partitioning of on-hand reasoning chains and (2) multi-level hints for problem solving.

In the following, we focus on detailing the latter two stages. We will frequently use the term "reasoning step" to refer to an intact unit of thought, which may consist of several sentences. Please note that this is distinct from a "training step," which refers to updating the model after processing a batch of data, or a "next-token prediction step," which generates a single token at a time.

#### 3.2.1 ADAPTIVE STEPWISE PARTITIONING OF ON-HAND REASONING CHAINS

**Definitions**   Let the reasoning chain be denoted as $G = t_1 \circ t_2 \circ \cdots \circ t_n$, where each $t_i$ represents a single token, and $\circ$ denotes concatenation. A *reasoning step* corresponds to a sequence of tokens $t_i \circ \cdots \circ t_j (1 \leq i \leq j \leq n)$ that forms an intact unit of thought. A *hint* is composed of one or more reasoning steps and serves as a conditioning prompt that guides the target model's reasoning toward a promising direction, helping it explore otherwise intractable solution spaces. The *level of a hint* is determined by the number of reasoning steps it contains—the more reasoning steps included, the richer the guidance it offers to the target model, and thus the higher its level.

**Method details**   We need a flexible method to adaptively partition a complete reasoning chain $G$ into $m$ reasoning steps, then combine appropriate number of reasoning steps as a hint in appropriate level to the target model. Conventional approach relies on syntactic cues, such as keywords like "first," "next," or "Step 1." However, such heuristics are prone to misidentifying the boundaries of reasoning steps and lack the flexibility. (Moens, 2017)

In contrast, we leverage the model's output probability distribution to identify the boundaries of reasoning steps. We hypothesize that when a reasoning step concludes, the model's probability of completing the entire chain should be relatively high. Conversely, at the beginning of a new step, this probability should decrease as the model expects additional reasoning to follow. This perceived likelihood of reasoning completion can be captured by the probability assigned to a special "end-of-thinking" token, `</think>`, which is explicitly introduced during pretraining to mark the conclusion of reasoning (Team, 2024; Guo et al., 2025). Formally, the model's tendency to conclude reasoning at token $t_i$ can be quantified by the probability $p(\text{</think>} \mid G_i)$, where $G_i$ denotes the token sequence up to $t_i$.

This hypothesis leads us to our core partitioning method: a token $t_i$ is considered a candidate reasoning step boundary if and only if the probability of concluding the reasoning chain after $t_i$ is greater than the probability of concluding it after the subsequent token, $t_{i+1}$: $p(\texttt{</think>}|G_i) > p(\texttt{</think>}|G_{i+1})$. By iterating through the entire reasoning chain, we collect all tokens satisfying this condition as candidate boundaries. To maintain high-quality reasoning steps, we enforce two constraints during partitioning: (1) adjacent boundaries must be at

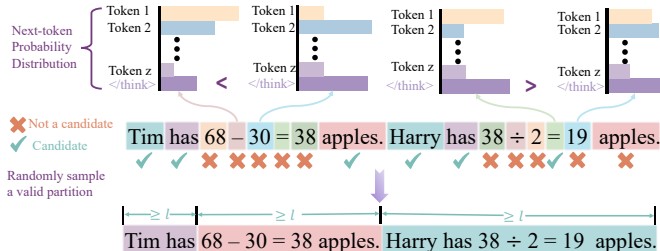

Figure 1: Adaptive stepwise partitioning of a reasoning chain: step boundaries are identified where the probability of concluding the reasoning chain after the current token, $p(\texttt{</think>}|G_i)$, is greater than concluding after the subsequent token, $p(\texttt{</think>}|G_{i+1})$. A final partition is chosen to meet constraints on step count $m$ and minimum length $l$.

least $l$ tokens apart to avoid overly short steps, and (2) the number of steps must equal the predetermined value, $m$. In practice, multiple valid partitionings may satisfy these constraints, we randomly sample one to proceed with. Figure 1 illustrates this token-distribution-based partitioning method. An alternative selection strategy and its effect is discussed in Appendix G.2.

### 3.2.2 Multi-level hints for problem solving

Building on the adaptive stepwise partitioning method described above, we divide the reasoning chain into $m$ reasoning steps. A key question is how many of these steps should be included as a complete hint for the target model.

An ideal-level hint matches the model's current capabilities—it is neither too weak nor too strong. Moreover, this optimal level shifts continuously as the model's reasoning ability evolves. Considering these challenges, rather than determining the ideal level at each training step, we provide hints at multiple levels. Our core hypothesis is that if the partitioning is fine-grained enough, there is likely to be a hint that fits the model's needs well. Specifically, we construct a set of $m - 1$ multi-level hints, $\mathcal{H}$. Each hint is a prefix of the full chain $G$, created by concatenating the first $j$ steps:

$$\mathcal{H} = \{h_j | h_j = S_1 \circ S_2 \circ \cdots \circ S_j, \quad \text{for } j = 1, \cdots, m - 1\},$$

where $S_i$ represents the $i$-th reasoning step. Low-level hints preserve considerable problem-solving difficulty. In contrast, high-level hints simplify the solution space, making the completion easier.

For each problem in the training set, we construct three types of prompts for the model to complete:

1. **Hinted problems:** For each of the $m - 1$ hint levels, the model is asked to complete the reasoning from that hint using $k_{\text{hint}}$ rollout attempts per hint.

2. **Unhinted problems:** To preserve the model's independent exploration, it also solves the problem from scratch without any hints. It is allowed $k_{\text{unhint}}$ rollouts in this case.

3. **Reference trajectory:** The original ground-truth reasoning chain $G$ is also provided to the target model and used to assign rewards. This ensures that the model is consistently exposed to the correct solution path during training.

Figure 2 illustrates this multi-level hinting and completion process. In total, the model generates $k_{\text{hint}}(m - 1) + k_{\text{unhint}} + 1$ completions per training problem, each receiving a reward based on correctness. StepHint strikes a balance between guiding the model with reliable hints and allowing it to learn from its own exploration mistakes, leading to more effective learning.

The above method applies to most RLVR algorithms but poses issues for GRPO (He et al., 2025), prompting further adaptations and discussion. In GRPO, an incorrect completion assigns negative advantages to all tokens in the rollout—including those in the correct hint prefix. This steers the model away from the correct reasoning chains. To address this, we modify GRPO by clipping

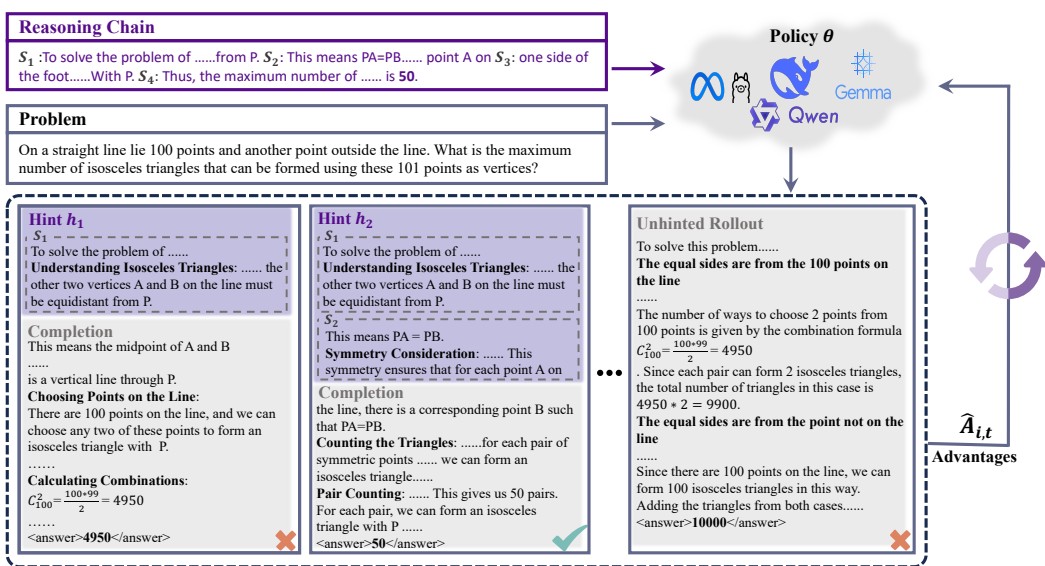

Figure 2: An overview of the multi-level hinting process. The process begins with a ground-truth reasoning chain, which is partitioned into $m$ steps (Section 3.2.1). From these steps, we construct $m-1$ prefix-based hints$(h_1, h_2, \cdots, h_{m-1})$. The model is trained to generate completions from each hint level, as well as from scratch (Unhinted), and a reference trajectory.

negative advantages to zero for tokens in the hint prefix when the completion is incorrect; that is, we set $\hat{A}_{i,t}^{\mathrm{GRPO}} = \max(0, \hat{A}_{i,t}^{\mathrm{GRPO}})$ (see Eq. 1 for the definition of $\hat{A}_{i,t}^{\mathrm{GRPO}}$). This adaptation prevents the model from being penalized for correct prefixes, aligning the training process with our intended mechanism. We empirically validate the effectiveness of this design in Appendix G.1 and prove the convergence of StepHint in Appendix H.

## 4 EXPERIMENTS

### 4.1 EXPERIMENTAL SETTINGS

**Training data**  To construct our training data, we generated complete reasoning chains for challenging problems. This process yield a training dataset of approximately 26,000 instances. The specific details of our data construction and processing methods are described in Appendix D.

**Hyperparameters**  We train two backbone models using GRPO (Shao et al., 2024) for 5 epochs each: Qwen-2.5-7B-Instruct (Yang et al., 2024a; Team, 2024) and Qwen-2.5-Math-7B (Yang et al., 2024b). The training prompt template is shown in Appendix C. The training is based on the VeRL framework (Sheng et al., 2024). Other training settings are detailed in Appendix E. The effect of applying an alternative optimization algorithm, Dr.GRPO, is further examined in Appendix G.3.

**Baselines**  We compare our method against five categories of baselines: (1) *Vanilla GRPO:* We train Qwen-2.5-7B-Instruct and Qwen-2.5-Math-7B using vanilla GRPO with the same settings as StepHint. (2) *Dr.GRPO:* We train Qwen-2.5-7B-Instruct using Dr.GRPO (Liu et al., 2025) under the same settings. (3) *SFT:* We fine-tune the backbone models with our dataset. (4) *Other RLVR Methods:* We evaluate several other reinforcement learning from verifier rewards (RLVR) enhancement techniques, including SimpleRL-Zero-7B (Zeng et al., 2025), Qwen-2.5-Math-7B-SimpleRL-Zoo, OpenReasonerZero-7B (Hu et al., 2025), and Oat-7B (Liu et al., 2025). (5) *RL with Reference Trajectory:* We include Luffy-Qwen-2.5-7B-Instruct (Yan et al., 2025) and Luffy-Qwen-Math-7B-Zero. For baselines in categories (4) and (5), we use their publicly released model weights and prompt templates for evaluation.

Table 1: Performance comparison of StepHint with baseline methods on in-domain and out-of-domain benchmarks. The top score in each column is in **bold**, and the second-highest is underlined. Backbone models are denoted by: [*]Qwen-2.5-7B-Instruct, [†]Qwen-2.5-7B, [‡]Qwen-2.5-Math-7B.

| Model | In-Domain | | | | | | | Out-of-Domain | | |
|---|---|---|---|---|---|---|---|---|---|---|
| | AIME24 (avg@5) | MATH500 (pass@1) | AMC (avg@5) | Olympiad (pass@1) | Minerva (pass@1) | AIME25 (avg@5) | Avg. - | ARC-C (pass@1) | GPQA-D (pass@1) | Avg. - |
| SFT[*] | 20.00 | 78.80 | 53.73 | 36.89 | 36.40 | 10.67 | 50.05 | 90.96 | 23.23 | 81.17 |
| SFT[‡] | 26.00 | 82.20 | 59.52 | 45.19 | 34.19 | 15.33 | 54.77 | 67.66 | 23.74 | 61.31 |
| **On-policy RLVR replication** | | | | | | | | | | |
| Vanilla-GRPO[*] | 24.67 | 76.60 | 51.33 | 43.41 | 39.34 | 10.67 | 52.59 | 91.30 | 37.37 | 83.51 |
| Vanilla-GRPO[‡] | 24.67 | 78.60 | 60.72 | 40.00 | 36.40 | 15.33 | 51.85 | 79.78 | 36.87 | 73.58 |
| Dr.GRPO[*] | 24.00 | 78.20 | 51.57 | 42.81 | 39.34 | 12.00 | 52.87 | 91.64 | 35.86 | 83.58 |
| **Other RLVR methods** | | | | | | | | | | |
| ORZ-7B[†] | 24.67 | 81.00 | 46.90 | 45.60 | 33.46 | 15.30 | 53.76 | 90.53 | 40.40 | 83.28 |
| SimpleRL[†] | 22.00 | 76.00 | 47.90 | 39.30 | 36.40 | 5.30 | 49.83 | 74.74 | 32.32 | 68.61 |
| SimpleRL[‡] | 28.00 | 76.20 | 57.59 | 37.93 | 34.93 | 12.00 | 49.80 | 63.91 | 27.27 | 58.61 |
| Oat[‡] | **36.00** | 78.40 | 59.75 | 42.52 | 36.40 | 10.00 | 52.92 | 59.89 | 33.84 | 56.13 |
| **RL with reference trajectory** | | | | | | | | | | |
| LUFFY[*] | 21.30 | 77.80 | 44.82 | 40.00 | 36.40 | 14.67 | 50.69 | 81.83 | 32.32 | 74.67 |
| LUFFY[‡] | 27.33 | 83.20 | 60.24 | 48.00 | 38.97 | 17.33 | 57.19 | 81.83 | 35.86 | 75.19 |
| **StepHint**[*] | 29.33 | 82.80 | 61.69 | 47.41 | **43.38** | 17.30 | 57.69 | **91.89** | **42.42** | **84.74** |
| **StepHint**[‡] | **36.00** | **87.00** | **62.65** | **52.15** | 38.24 | **18.87** | **60.35** | 84.73 | 38.89 | 78.10 |

**Evaluation** We follow prior work (Yan et al., 2025) and evaluate on six math datasets: AIME 2024, AIME 2025, AMC (Li et al., 2024), Minerva (Lewkowycz et al., 2022), OlympiadBench (He et al., 2024), and MATH500 (Hendrycks et al., 2021). For the AIME 24/25 and AMC datasets, given their limited data points, we conduct each evaluation five times and report the average results. To evaluate generalization, we also incorporate two non-math benchmarks, ARC-C (Clark et al., 2018) and GPQA-D (Rein et al., 2024), as out-of-domain tests. We report the weighted average accuracy for both in-domain and out-of-domain benchmarks. The generation length is also set to 4,110. All results were evaluated using Math-Verify[2] and OAT-Grade (Liu et al., 2024) on 8×A100s.

## 4.2 MAIN RESULTS

Table 1 shows the overall performance of StepHint and baseline methods.

When applied to the general-purpose model, Qwen-2.5-7B-Instruct, StepHint achieves the highest performance on in-domain math tasks among all compared methods. Compared to other RLVR methods, StepHint shows a 3.93 percentage point improvement over the next-best method, ORZ. Furthermore, StepHint consistently surpasses the SFT baseline, indicating that StepHint effectively learns beyond simple token imitation, leading to improved reasoning outcomes.

Notably, the Qwen-2.5-7B-Instruct model trained with StepHint outperforms the specialized Qwen-2.5-Math-7B trained with any other method, highlighting the substantial boost in reasoning ability provided by StepHint and allowing a generalist model to outperform a specialist in its own domain.

For the specialized Qwen-2.5-Math-7B model, as expected from its specialized design (Yang et al., 2024b), the Math model performs lower on the out-of-domain benchmarks compared to the general-purpose Qwen-2.5-7B-Instruct. However, StepHint not only leads the board in in-domain math tasks compared with baselines but also enables the Math model to achieve the highest out-of-domain test performance among all baselines. This suggests that the improvements are not solely due to domain-specific knowledge but also reflect an enhancement of the model's general reasoning capabilities.

---

[2]https://github.com/huggingface/Math-Verify

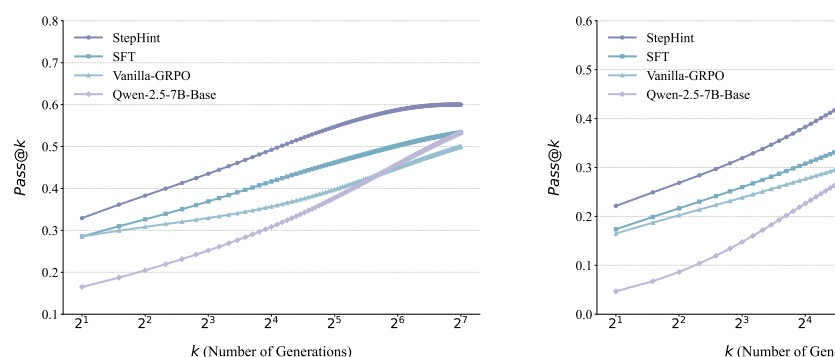

Figure 3: Comparison of pass@k results on the AIME24 and AIME25 datasets. **Left:** AIME24; **Right:** AIME25.

We now turn our analysis to Luffy, an RL method that employs an *entire* reasoning chain from a stronger model as a reference trajectory. A key limitation of this paradigm is its potential inability to fully address the near-miss reward issue, as the model's exploration is not directly guided. Although using expert trajectories can mitigate exploration stagnation, it may come at the expense of the intrinsic exploration mechanism vital to reinforcement learning. In contrast, StepHint integrates external hints with the model's own exploration, fostering a more robust learning process. Our experimental results substantiate this claim, demonstrating that StepHint outperforms Luffy on both in-domain and out-of-domain tasks.

To further assess the generalizability of our approach, we conducted experiments on Llama-3.1-8B (Dubey et al., 2024), comparing StepHint with GRPO. The results, presented in Table 2, demonstrate that our method achieves superior performance. Training settings are detailed in Appendix F.

Table 2: Results on Llama-3.1-8B.

|  | **AIME24** | **MATH500** | **AMC** | **Olympiad** | **Minerva** | **AIME25** | **Avg.** |
|---|---|---|---|---|---|---|---|
| *GRPO* | 4.00 | 9.00 | 6.02 | 3.56 | 12.13 | 0 | 6.81 |
| *StepHint* | 6.00 | 44.00 | 18.56 | 13.48 | 20.22 | 1.30 | 24.13 |

### 4.3 PASS@K EVALUATION

Many studies (Chen et al., 2021; Wang et al., 2022) show that with a limited number of rollouts, models may perform poorly on certain tasks. However, with a sufficiently large number of rollouts, they are more likely to solve these problems. Therefore, to fully assess the model's potential performance, pass@k accuracy is a more suitable metric (Yue et al., 2025). In this context, a problem is considered solved if any of the $k$ sampled reasoning chains yields a correct answer.

Figure 3 presents the pass@k results on the AIME24 and AIME25 datasets. The results demonstrate that StepHint improves the model's pass@k performance. In contrast, Vanilla-GRPO shows a slower rate of improvement at higher values of $k$, which aligns with findings from previous work (Yue et al., 2025). The superior performance under pass@k evaluation further validates the effectiveness of StepHint. Additionally, the performance difference can be attributed to the exploration strategies the models employ. While Vanilla-GRPO suffers from "exploration stagnation," StepHint guides the model's exploration, helping it break free from its "comfort zone."

### 4.4 METHOD ANALYSIS FROM TRAINING DYNAMICS

Figure 4 illustrates the training dynamics of StepHint and Vanilla-GRPO on Qwen-2.5-7B-Instruct, comparing them across three key metrics: entropy, response length, and training rewards.

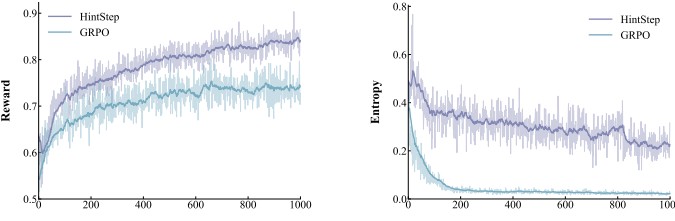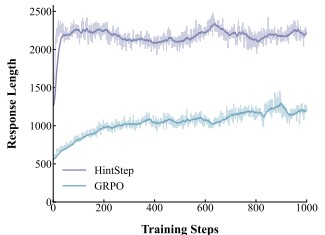

Figure 4: Training dynamics of StepHint compared with GRPO. **Left:** Reward; **Middle:** Entropy; **Right:** Response Length.

**Reward**    The reward curves highlight the different learning phases. Due to the multi-level step-wise hints provided by StepHint, the problem-solving difficulty for the model is lower compared to Vanilla-GRPO. As a result, the reward score for StepHint is consistently higher, reflecting the mitigation of the near-miss reward issue. A closer examination of the trends in both curves offers further insights. Vanilla-GRPO shows a steady, monotonic increase in reward as it refines its existing policy. In contrast, StepHint experiences a brief initial dip in reward before a rapid and sustained increase. This initial dip likely reflects an adaptation period where the model transitions from simple exploitation to a more complex, hint-guided exploration. Once adapted, the model efficiently discovers higher-reward solutions, leading to faster and effective learning to reason.

**Entropy**    Both methods exhibit an overall decrease in entropy, though their trajectories diverge as training progresses. The entropy of StepHint remains higher than that of Vanilla-GRPO. This suggests that the hints provided by StepHint encourage a more diverse policy, preventing premature convergence to a narrow solution subspace and promoting a higher level of exploration (Cheng et al., 2025). This trend reflects, to some extent, the mitigation of exploration stagnation.

**Response length**    The two methods demonstrate distinct patterns in generated response length. StepHint shows an initial sharp increase in length, which we attribute to the model learning to mimic the structured, stepwise reasoning chains provided by the multi-level hints. These hints are often more detailed than the model's initial, more direct responses.

## 5 RELATED WORKS

RL-based post-training has demonstrated remarkable success in mathematical reasoning tasks (Shao et al., 2024; Yang et al., 2024b). Research in this area has primarily advanced in three directions: (1) optimizing the models and their training data, (2) refining inference-time strategies, and (3) improving policy optimization methods.

The first direction involves constructing high-quality, large-scale mathematical reasoning datasets (Wang et al., 2023; Ye et al., 2025; Zhao et al., 2025) and designing specialized training or fine-tuning methods (Jaech et al., 2024; Mitra et al., 2024). The second direction focuses on guiding the model's step-by-step thought processes without altering its underlying weights, typically through sophisticated prompting techniques such as Chain-of-Thought (Wei et al., 2022) and innovations in in-context learning (Wu et al., 2024). The third direction aims at developing advanced policy optimization algorithms. GRPO, an advancement of PPO (Schulman et al., 2017), has recently gained popularity due to its simplicity and strong performance (Hu et al., 2025; Zeng et al., 2025). Additionally, several improvements have been proposed for GRPO; for example, Liu et al. (2025) identified inherent length and difficulty biases in vanilla GRPO and addressed these issues.

## 6 CONCLUSION

In this paper, we introduced StepHint, a novel RLVR algorithm that incorporates multi-level step-wise hints. This mechanism is designed to provide the model with assistance tailored to its evolving capabilities, thereby facilitating the learning process by addressing challenges such as near-miss re-wards and exploration stagnation. StepHint not only outperforms strong baselines on mathematical benchmarks but also demonstrates robust generalization to out-of-domain tasks, highlighting the promising potential of the stepwise hinting paradigm for RLVR enhancement.

## REPRODUCIBILITY STATEMENT

The code used for training StepHint is available at `https://anonymous.4open.science/r/StepHint-AC69`. We will release the model weights at a later date.

Training Data: The models and methodology for constructing our training dataset is detailed in Appendix D.

Training Details: The specifics of the training process are described in Section 4.1 and Appendix E.

Evaluation Details: The hyperparameters used for evaluation are also detailed in Section 4.1.

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

## A    USE OF LLMS

As described in Appendix E, we use LLMs to generate high-quality reasoning chains and partition these chains into logical steps. We believe that this use of LLMs serves as a justified auxiliary tool, and this controlled process does not introduce academic or ethical risks.

## B    PROOF OF PROPOSITION 1

Proposition 1:
$$\mathbb{E}_{t_k \sim p(\cdot|S_{k-1})}[H(\mathcal{R}|S_k)] \leq H(\mathcal{R}|S_{k-1})$$

*Proof.* We want to prove the following inequality:
$$\mathbb{E}_{t_k \sim p(\cdot|S_{k-1})}[H(\mathcal{R}|S_k)] \leq H(\mathcal{R}|S_{k-1})$$

This inequality states that the expected entropy of solution space $\mathcal{R}$ conditioned on the state $S_k$ is less than or equal to the entropy of $\mathcal{R}$ conditioned on the prior state $S_{k-1}$. The state $S_k$ is reached from $S_{k-1}$ after an observation or transition $t_k$. Let's denote the random variable for this transition as $T_k$.

First, let's clarify the notation. The expression on the left-hand side, $\mathbb{E}_{t_k \sim p(\cdot|S_{k-1})}[H(\mathcal{R}|S_k)]$, represents the conditional entropy $H(\mathcal{R}|S_k)$. The conditional entropy $H(X|Y)$ is defined as the expectation of the entropy of $X$ over the values of $Y$. The subscript simply makes the underlying probability model explicit: the distribution of the state $S_k$ is induced by the distribution of the prior state $S_{k-1}$ and the transition $T_k$.

The proof of this relationship relies on the non-negativity of conditional mutual information. The conditional mutual information between two random variables, $\mathcal{R}$ and $T_k$, given a third variable $S_{k-1}$, is defined as:

$$I(\mathcal{R}; T_k | S_{k-1}) = H(\mathcal{R}|S_{k-1}) - H(\mathcal{R}|S_{k-1}, T_k)$$

A fundamental property of mutual information is that it is always non-negative (MacKay, 2003; Polyanskiy & Wu, 2025):

$$I(\mathcal{R}; T_k | S_{k-1}) \geq 0$$

From this, it directly follows that:

$$H(\mathcal{R}|S_{k-1}) \geq H(\mathcal{R}|S_{k-1}, T_k)$$

This equation shows that conditioning on an additional variable, $T_k$, can only decrease (or leave unchanged) the entropy of $\mathcal{R}$.

Now, we must relate the term $H(\mathcal{R}|S_{k-1}, T_k)$ to the term $H(\mathcal{R}|S_k)$. The state $S_k$ is the result of a process that starts in state $S_{k-1}$ and undergoes the transition $T_k$. This means that the information contained in the pair of variables $(S_{k-1}, T_k)$ fully determines the state $S_k$. In many typical models, knowing $S_k$ is equivalent to knowing the pair $(S_{k-1}, T_k)$ that produced it. If we assume this equivalence, then conditioning on $S_k$ is the same as conditioning on the pair $(S_{k-1}, T_k)$. Therefore, we have:

$$H(\mathcal{R}|S_k) = H(\mathcal{R}|S_{k-1}, T_k)$$

Substituting this equality back into our previous inequality, we arrive at the desired result:

$$H(\mathcal{R}|S_k) \leq H(\mathcal{R}|S_{k-1})$$

This completes the proof. □

## C  TEMPLATE OF QWEN-2.5

---
**Template of Qwen-2.5**

`<|im_start|>`system
You are a helpful assistant. The assistant first thinks about the reasoning process in the mind and then provides the user with the answer. The reasoning process and answer are enclosed within `<think>` `</think>` and `<answer>` `</answer>` tags, respectively, i.e., `<think>` reasoning process here `</think><answer>` answer here `</answer>`. Now the user asks you to solve a mathematical reasoning problem. After thinking, when you finally reach a solution, clearly state the answer marked with \boxed{} and within `<answer>` `</answer>` tags, i.e., `<answer>`\boxed{answer}`</answer>`
`<|im_end|>`
`<|im_start|>` user
{question}
`<|im_end|>`
`<|im_start|>`assistant
`<think>`

---

## D  TRAINING DATA CONSTRUCTION

We gathered all problems from the DAPO dataset (Yu et al., 2025) and selected problems of difficulty level 7 or higher from the DEEPMATH dataset (He et al., 2025). The DEEPMATH dataset provides solution reasoning chains, while for each question in the DAPO dataset (Yu et al., 2025), we sample a total of 12 reasoning chains using DAPO-Qwen-32B (Yu et al., 2025), QWQ-32B (Team, 2025), and DeepSeek-R1-Distill-Qwen-32B (Guo et al., 2025), with 4 samples per model. The sampling is conducted under a 0-shot setting, with a temperature of 1 and a maximum length of 8,192. We filter these to retain all reasoning chains that are both correct and have a length of no more than 4,110. In cases where multiple chains satisfy these conditions, we randomly select one.

All reasoning chains are partitioned into $m = 4$ steps with the method we proposed in Section 3.2.1, each longer than $l = \frac{L}{8}$ tokens, where $L$ is the total length of a reasoning chain $G$, with QWQ-32B (Team, 2025). This threshold ensures that even the shortest step contains a substantive amount of information, while allowing for the natural length variation between different steps in a complex reasoning process.

## E  TRAINING SETUP

We set a global batch size of 128 and a fixed learning rate of $1e-6$. Following (Yan et al., 2025), we set the KL loss coefficient $\beta = 0$, indicating no reference model is used for regularization. We configure $k_{hint} = 2$ and $k_{unhint} = 5$. During training, the temperature for rollout generation is set to 1.0. Our training is completed on $8\times$A100s.

Since the Qwen-2.5-Math-7B model (Yang et al., 2024b) has a relatively short context length of 4,096 tokens, we adopt a community-released variant that extends the context length to 32k tokens.[3]

## F  TRAINING SETTINGS FOR LLAMA

Due to temporal and computational constraints, we train the Llama-3.1-8B model for 4 epochs on our dataset using two methods: StepHint and GRPO. We increased the maximum generation length to 4,500 during training to ensure it could accommodate the longest external reasoning chains within the dataset. To align with the original Llama pre-training paradigm, we introduced slight modifications to the prompt template. All other hyperparameters remained consistent with those of the Qwen-2.5 models.

---

**Template of Llama-3.1**

system

You are a helpful assistant. The assistant first thinks about the reasoning process in the mind and then provides the user with the answer. The reasoning process and answer are enclosed within `<think> </think>` and `<answer> </answer>` tags, respectively, i.e., `<think>` reasoning process here `</think><answer>` answer here `</answer>`. Now the user asks you to solve a mathematical reasoning problem. After thinking, when you finally reach a solution, clearly state the answer marked with \boxed{} and within `<answer> </answer>` tags, i.e., `<answer>`\boxed{answer}`</answer>`

user

{question}

assistant

`<think>`

---

## G  ABLAION STUDY

### G.1  NON-NEGATIVE ADVANTAGE FOR HINTS

We perform an ablation study with the Qwen2.5-7B-Instruct model to assess the effect of enforcing a non-negative constraint on the advantages of hints. As shown in Table 3, *w/ Constraint* denotes the configuration where the advantage values of hints are constrained to be non-negative, whereas *baseline* corresponds to the baseline without this constraint. The results demonstrate that introducing the non-negative constraint improves the model's mathematical reasoning performance.

### G.2  STEPWISE PARTITIONING STRATEGY

In this section, we conduct an ablation study on the stepwise partitioning strategy introduced in Section 3.2.1. We compare two approaches: *Base*, which selects $k$ tokens uniformly at random from

---

[3] `https://huggingface.co/open-r1/Qwen2.5-Math-7B-RoPE-300k`

Table 3: Ablation results on the effect of applying non-negative advantage constraint for hints.

| | AIME24 | MATH500 | AMC | Olympiad | Minerva | AIME25 | Avg. |
|---|---|---|---|---|---|---|---|
| | *Qwen2.5-7B-Instruct-StepHint* | | | | | | |
| *baseline* | 28.00 | 81.40 | 57.30 | 45.30 | 39.70 | 14.70 | 55.43 |
| *w/ Constraint* | 29.33 | 82.80 | 61.69 | 47.41 | 43.38 | 17.30 | 57.69 |

candidate tokens that satisfy the condition $(p(\texttt{</think>}|G_i) > p(\texttt{</think>}|G_{i+1}))$ and the interval constraints; and *Salient*, which selects the top-$k$ candidate tokens exhibiting the largest probability drop $(p(\texttt{</think>}|G_i) - p(\texttt{</think>}|G_{i+1}))$ while satisfying the interval constraints.

Table 4: Performance comparison of different stepwise partitioning strategies.

| Strategy | AIME24 | MATH500 | AMC | Olympiad | Minerva | AIME25 | Avg. |
|---|---|---|---|---|---|---|---|
| | *Qwen2.5-7B-Instruct-StepHint* | | | | | | |
| *Base* | 29.33 | 82.80 | 61.69 | 47.41 | 43.38 | 17.30 | 57.69 |
| *Salient* | 29.33 | 83.80 | 59.28 | 47.41 | 43.01 | 18.67 | 57.84 |
| | *Qwen2.5-Math-7B-StepHint* | | | | | | |
| *Base* | 36.00 | 87.00 | 62.65 | 52.15 | 38.24 | 18.87 | 60.35 |
| *Salient* | 36.00 | 86.60 | 63.86 | 53.33 | 38.24 | 18.67 | 60.78 |

As shown in Table 4, both strategies achieve comparable overall performance.

### G.3 OPTIMIZATION ALGORITHM

In this section, we evaluate the effect of different optimization algorithms on our proposed StepHint method. Specifically, we employ Dr.GRPO as the optimization algorithm and compare its performance against Vanilla-GRPO. The results are summarized in Table 5.

Table 5: Performance comparison of StepHint under different optimization algorithms.

| | AIME24 | MATH500 | AMC | Olympiad | Minerva | AIME25 | Avg. |
|---|---|---|---|---|---|---|---|
| | *Qwen2.5-7B-Instruct-StepHint* | | | | | | |
| *GRPO* | 29.33 | 82.80 | 61.69 | 47.41 | 43.38 | 17.30 | 57.69 |
| *Dr.GRPO* | 32.00 | 82.40 | 61.00 | 49.04 | 42.65 | 18.0 | 58.15 |
| | *Qwen2.5-Math-7B-StepHint* | | | | | | |
| *GRPO* | 36.00 | 87.00 | 62.65 | 52.15 | 38.24 | 18.87 | 60.35 |
| *Dr.GRPO* | 40.67 | 88.60 | 65.30 | 51.70 | 40.44 | 22.67 | 61.33 |

## H PROOF OF CONVERGENCE

We first recall two standard assumptions from stochastic non-convex optimization.

**Assumption 1** (Smoothness). *The objective function $J(\theta)$ is continuously differentiable, and its gradient is L-Lipschitz:*

$$\|\nabla J(\theta_1) - \nabla J(\theta_2)\| \leq L\|\theta_1 - \theta_2\|, \quad \forall \theta_1, \theta_2.$$

*Consequently, for any update $\theta_{t+1}$:*

$$J(\theta_{t+1}) \geq J(\theta_t) + \langle \nabla J(\theta_t), \theta_{t+1} - \theta_t \rangle - \frac{L}{2}\|\theta_{t+1} - \theta_t\|^2.$$

**Assumption 2** (Bounded Variance). *The variance of the stochastic gradient estimator $\hat{g}_t$ is bounded:*

$$\mathbb{E}\big[\|\hat{g}_t - \mathbb{E}[\hat{g}_t|\theta_t]\|^2 \,\big|\, \theta_t\big] \leq \sigma^2.$$

**Proposition 2.** *Let $\hat{g}_t$ denote the token-averaged PPO/GRPO stochastic gradient estimator used by StepHint at iteration $t$. Let $p_{hint} \in [0,1]$ be the expected fraction of tokens in a rollout that are hint tokens, and let $\alpha \in [0,1]$ be the probability that a rollout produced under the current policy yields a correct final outcome. Assume the following modeling approximations at iteration $t$:*

*(A1)* ***Token homogeneity:*** *the expected per-token policy-gradient contribution within a rollout is approximately the same across token positions, i.e.*

$$\mathbb{E}\big[\nabla_\theta \log \pi_\theta(\tau_{i,j})\hat{A}_{i,j}\big] \approx c_t\, \nabla J(\theta_t)$$

*for some scalar $c_t > 0$, so that token-level expectations align with the full policy gradient direction.*

*(A2)* ***Clipping/ratio approximation for hint tokens:*** *for hint tokens in rollouts that produce a correct outcome the PPO probability ratio $r_{i,j} = \pi_\theta(\tau_{i,j})/\pi_{old}(\tau_{i,j})$ is typically above the upper clip $1 + \epsilon$, hence the clipped surrogate multiplies the advantage by approximately $1 + \epsilon$; for non-hint tokens the ratio is approximated as $r_{i,j} \approx 1$ so clipping is inactive.*

*Under (A1)–(A2) define*

$$\beta_t := (1 - p_{hint}) + \alpha\, p_{hint}(1 + \epsilon).$$

*Then the conditional expectation of the StepHint gradient estimator can be approximated by*

$$\mathbb{E}[\hat{g}_t \mid \theta_t] \approx \beta_t\, \nabla J(\theta_t).$$

*Proof.* Write $\hat{g}_t$ for the token-averaged stochastic gradient estimator at iteration $t$. Let a rollout contain multiple tokens and let $g_j$ denote the gradient contribution (surrogate objective / policy-gradient contribution) associated with a generic token position $j$ in a rollout. We drop the rollout index for notational simplicity and condition all expectations on $\theta_t$.

The token-averaged estimator can be written as an expectation over token positions:

$$\mathbb{E}[\hat{g}_t \mid \theta_t] = \mathbb{E}[g_j \mid \theta_t] = (1 - p_{hint})\,\mathbb{E}[g_j \mid \text{token } j \text{ is not a hint}, \theta_t] + p_{hint}\,\mathbb{E}[g_j \mid \text{token } j \text{ is a hint}, \theta_t],$$

since a token is either a hint token (fraction $p_{hint}$) or a non-hint token (fraction $1 - p_{hint}$).

We treat the two terms separately.

**Non-hint tokens.** For non-hint tokens the surrogate is the standard PPO/GRPO surrogate built from the student policy's own probabilities and advantages. Under Assumption (A1) we approximate the per-token expected contribution by a common direction proportional to the full policy gradient:

$$\mathbb{E}[g_j \mid \text{non-hint}, \theta_t] \approx \mathbb{E}[\nabla_\theta \log \pi_\theta(\tau)\, \hat{A}],$$

**Hint tokens.** For tokens that are provided as hints we further condition on whether the *entire* rollout ends in a correct final outcome. Let $\alpha$ denote the probability (under the current policy and the environment) that a rollout is correct. Then, conditioning on hint token and rollout correctness,

$$\mathbb{E}[g_j \mid \text{hint}, \theta_t] = (1 - \alpha)\,\mathbb{E}[g_j \mid \text{hint}, \text{rollout incorrect}, \theta_t] + \alpha\,\mathbb{E}[g_j \mid \text{hint}, \text{rollout correct}, \theta_t].$$

By the StepHint design (and the modification to GRPO described in the paper), when a rollout is incorrect the negative advantages assigned to hint tokens are clipped to zero (i.e. the algorithm

prevents penalizing the model for hint tokens in incorrect rollouts). Therefore the contribution from hint tokens in incorrect rollouts is (approximately) zero:

$$\mathbb{E}[g_j \mid \text{hint}, \text{rollout incorrect}, \theta_t] \approx 0.$$

For hint tokens in *correct* rollouts, Assumption (A2) posits that the PPO probability ratio $r_j = \frac{\pi_\theta(\tau_j)}{\pi_{old}(\tau_j)}$ is typically above the upper clipping threshold $1 + \epsilon$, so that the clipped surrogate evaluates approximately to $(1 + \epsilon)\hat{A}$. Together with the token-homogeneity approximation (A1) that aligns per-token expectation with $G_t$, we obtain

$$\mathbb{E}[g_j \mid \text{hint}, \text{rollout correct}, \theta_t] \approx (1 + \epsilon)\, G_t.$$

Combining the two subcases for hint tokens gives

$$\mathbb{E}[g_j \mid \text{hint}, \theta_t] \approx (1 - \alpha) \cdot 0 \; + \; \alpha \cdot (1 + \epsilon)\, G_t = \alpha(1 + \epsilon)\, G_t.$$

**Combine hint and non-hint contributions.** Substitute the approximations for the two conditional expectations back into the decomposition at the top:

$$\mathbb{E}[\hat{g}_t \mid \theta_t] \approx (1 - p_{hint})\, G_t \; + \; p_{hint}\big(\alpha(1 + \epsilon)\, G_t\big) = \big((1 - p_{hint}) + \alpha p_{hint}(1 + \epsilon)\big)\, G_t.$$

Recalling $G_t \equiv \mathbb{E}[\nabla_\theta \log \pi_\theta(\tau)\, \hat{A}] = \nabla J(\theta_t)$ (the standard policy-gradient direction under our estimator), we obtain

$$\mathbb{E}[\hat{g}_t \mid \theta_t] \approx \beta_t \nabla J(\theta_t), \qquad \text{where} \quad \beta_t := (1 - p_{hint}) + \alpha p_{hint}(1 + \epsilon).$$

This completes the derivation. $\qquad\qquad\qquad\qquad\qquad\qquad\qquad\qquad\qquad\qquad\qquad\qquad\square$

**Theorem H.1** (Convergence to stationarity). *Suppose Assumptions 1–2 hold and that Proposition 2's approximation is valid with $\beta_t \in [\beta_{\min}, \beta_{\max}]$ for constants $0 < \beta_{\min} \leq \beta_{\max} < \infty$. Consider stochastic ascent updates*

$$\theta_{t+1} = \theta_t + \eta \hat{g}_t$$

*with fixed step size satisfying*

$$\eta \leq \frac{1}{L\, \beta_{\max}}.$$

*Let $J^* = \sup_\theta J(\theta) < \infty$. Then for every integer $T \geq 1$,*

$$\frac{1}{T} \sum_{t=0}^{T-1} \mathbb{E}\big\|\nabla J(\theta_t)\big\|^2 \leq \frac{2\,(J^* - J(\theta_0))}{\eta T \beta_{\min}} + \frac{L\,\eta\,\sigma^2}{\beta_{\min}}.$$

*Hence, choosing $\eta = \Theta(1/\sqrt{T})$ yields the standard stochastic non-convex rate*

$$\min_{0 \leq t < T} \mathbb{E}\big\|\nabla J(\theta_t)\big\|^2 = O\big(T^{-1/2}\big),$$

*i.e., the iterates converge to a stationary point in expectation at the usual $O(1/\sqrt{T})$ speed.*

*Proof.* Starting from Assumption 1 and substituting $\Delta = \theta_{t+1} - \theta_t = \eta \hat{g}_t$ gives

$$J(\theta_{t+1}) \geq J(\theta_t) + \eta\langle\nabla J(\theta_t), \hat{g}_t\rangle - \tfrac{L\eta^2}{2}\|\hat{g}_t\|^2.$$

Take the conditional expectation $\mathbb{E}_t[\cdot] = \mathbb{E}[\cdot \mid \theta_t]$ and apply Proposition 2 to replace $\mathbb{E}_t[\hat{g}_t]$ by $\beta_t \nabla J(\theta_t)$ (approximately):

$$\mathbb{E}_t[J(\theta_{t+1})] \geq J(\theta_t) + \eta\beta_t\|\nabla J(\theta_t)\|^2 - \tfrac{L\eta^2}{2}\mathbb{E}_t\|\hat{g}_t\|^2.$$

Use the variance decomposition $\mathbb{E}_t\|\hat{g}_t\|^2 = \|\mathbb{E}_t[\hat{g}_t]\|^2 + \mathbb{E}_t\|\hat{g}_t - \mathbb{E}_t[\hat{g}_t]\|^2$ together with Assumption 2 to obtain

$$\mathbb{E}_t\|\hat{g}_t\|^2 \leq \beta_t^2\|\nabla J(\theta_t)\|^2 + \sigma^2.$$

Substituting back,

$$\mathbb{E}_t[J(\theta_{t+1})] - J(\theta_t) \geq \left(\eta\beta_t - \tfrac{L\eta^2\beta_t^2}{2}\right)\|\nabla J(\theta_t)\|^2 - \tfrac{L\eta^2\sigma^2}{2}.$$

With the choice $\eta \le 1/(L\beta_{\max})$ we have $L\eta\beta_t \le 1$, hence

$$\eta\beta_t - \frac{L\eta^2\beta_t^2}{2} = \eta\beta_t\left(1 - \frac{L\eta\beta_t}{2}\right) \ge \frac{\eta\beta_t}{2}.$$

Thus

$$\frac{\eta\beta_t}{2}\|\nabla J(\theta_t)\|^2 \le \mathbb{E}_t[J(\theta_{t+1}) - J(\theta_t)] + \frac{L\eta^2\sigma^2}{2}.$$

Taking total expectation, summing $t = 0, \ldots, T-1$, and using the bound $\beta_t \ge \beta_{\min} > 0$ yields

$$\frac{\eta\beta_{\min}}{2}\sum_{t=0}^{T-1}\mathbb{E}\|\nabla J(\theta_t)\|^2 \le \mathbb{E}[J(\theta_T)] - J(\theta_0) + \frac{TL\eta^2\sigma^2}{2}.$$

Since $\mathbb{E}[J(\theta_T)] \le J^*$ we obtain

$$\frac{1}{T}\sum_{t=0}^{T-1}\mathbb{E}\|\nabla J(\theta_t)\|^2 \le \frac{2(J^* - J(\theta_0))}{\eta T\beta_{\min}} + \frac{L\eta\sigma^2}{\beta_{\min}},$$

which proves the theorem. Choosing $\eta \propto T^{-1/2}$ gives the stated $O(1/\sqrt{T})$ rate. $\qquad\square$

