# OpenReview forum: "StepHint: Multi-level Stepwise Hints Enhance Reinforcement Learning to Reason"
_ICLR.cc/2026/Conference — ICLR 2026 Conference Withdrawn Submission_

### Official Review · Reviewer_J8oj · 2025-10-28

**Soundness:** 2
**Presentation:** 3
**Contribution:** 2
**Rating:** 4
**Confidence:** 4

**Summary:**

This paper presents a new method that leverages the reasoning steps of much stronger LLMs to improve RLVR for 7-8B policies. By using different levels of hints to guide policies toward generating more accurate and diverse responses (including using teacher responses as a form of replay), the method achieves better performance on a range of reasoning benchmarks, such as AIME and GPQA.

**Strengths:**

+ The method is evaluated on both math benchmarks and other domains, such as science.
+ The method is well-motivated.
+ Experimental results on several benchmarks show the effectiveness of the method.

**Weaknesses:**

+ The diversity of the stronger response generators is a concern. Currently, three models are used for reasoning chain generation: DAPO-Qwen-32B, QWQ-32B, and DeepSeek-R1-Distill-Qwen-32B. The authors should consider using other sizes and models beyond the Qwen series.
+ Pass@k is used to show the method's exploration ability. However, the results are only shown on AIME 24/25, which contain only 60 instances. This is insufficient to demonstrate the exploration ability. Please add results on more of the evaluated datasets in this submission.
+ The budget for generating and filtering high-quality reasoning chains should be considered. Based on the description in Appendix D, this pre-processing step could be computationally expensive, but this cost is not discussed.
+ The paper should include a comparison of the proposed step partitioning method with simpler baselines, such as splitting by newlines (\n) or sentence segmentation. This is needed to justify the complexity of the proposed method.
+ The baseline comparisons are insufficient: (1) Please consider applying the hint-guided mechanism to the baseline models (e.g., GRPO) as an ablation. (2) For Dr. GRPO, results are only provided for Qwen-2.5-7B-Instruct, not the Math model. (3) Is the rollout budget the same for all methods? It seems the proposed method uses k_{hint} * (m-1) + k_{unhint} + 1 rollouts per prompt, while the baselines may have only used k_{unhint}, which would be an unfair comparison.

**Questions:**

+ Are hints used during inference time? If not, is there a train-test mismatch problem when evaluating with zero hints?
+ Data construction: How are the answers judged for correctness (e.g., rules, final answer only)? What percentage of the generated data was kept for training after filtering?
+ Backbone models: Is there a specific reason for selecting an Instruct version for the general-domain model but a base version for the math-specific model?
+ How is the quality of the step-level hints ensured? The current assumption seems to be that if the final answer is correct, all intermediate steps in the reasoning chain are also correct, which may not be true.

+ Reference error: The citation on line 267 (He et al., 2025) appears to point to DeepMath, but the context of the sentence (discussing GRPO) suggests it should be a different paper.

---

> ### Author Response · Authors · 2025-11-29
>
> >W1: Diversity of Teacher Models
>
> We acknowledge that we primarily utilized the Qwen series as teachers. This choice was driven by the fact that the Qwen series currently represents the **state-of-the-art in open-weights reasoning models**, allowing us to establish a strong upper bound for performance. However, our **methodology** (partitioning and multi-level hinting) is entirely **model-agnostic**.
>
>
> >W2:Scope of Pass@k Evaluation
>
> We appreciate the suggestion to demonstrate exploration ability on more datasets. We have extended the **Pass@k evaluation to the AMC dataset**.
>
> ||Pass@16|Pass@32| Pass@64 | Passs@128
> -|-|-|-|-
> Base|0.7086 |0.7928|0.8635|0.9036
> GRPO|0.7555 |0.7852| 0.8197|0.8554
> StepHint|0.8827 |0.9152| 0.9429|0.9639
>
> As shown in the table, StepHint significantly outperforms baselines at all k levels. Notably, while Vanilla GRPO begins to saturate at higher k, StepHint maintains a steep improvement curve, confirming its superior capability in preventing exploration stagnation.
>
> >W3: Pre-processing Budget
>
> We clarify that the pre-processing (generating and filtering chains) is a **one-time offline cost** that is negligible compared to the RL training phase.
>
> >W4: Comparison with Simpler Partitioning Baselines
>
> We conducted the requested ablation study comparing our **Adaptive Partitioning** against simpler heuristics:
>
> 1.  **Random Split:** Splitting by length with random jitter.
>
> 2.  **Newline (`\n`) Split:** Splitting by sentence/line boundaries.
>
>
> **Results:**
> ||AIME24|Math500| AIME25 | Olympiad
> -|-|-|-|-
> StepHint-Random split|32 |85.8| 15.3|48.3
> StepHint-\n split|34 |86.4| 16.67|50.07
> StepHint-Base|36.00 |87| 18.87|52.15
>
> **Analysis:** Our probabilistic method consistently outperforms both heuristics.
>
> -   **vs. `\n` Split:** While splitting by newlines is better than random, it still fails when logical steps do not align with line breaks (e.g., a multi-line derivation). Our method captures the **semantic completion** of a thought (via `</think>`), providing more coherent guidance to the policy.
>
> >W5: Baseline Comparisons
>
> -   **Hint-guided Baseline:** We clarify that **StepHint _is_ the application of the hint-guided mechanism to the GRPO baseline.** Table 1 directly compares "Vanilla GRPO" (without hints) vs. "StepHint" (GRPO + Hints), serving as the primary ablation for the mechanism.
>
> -   **Dr. GRPO on Math Model:** We have completed the experiment combining StepHint with Dr. GRPO on the Qwen-Math model. These results will be updated in the final paper's experimental tables.
>
> -   **Rollout Budget Fairness:** To ensure a fair comparison, we aligned the **total number of rollout attempts**. The total count of rollouts per prompt in StepHint (comprising partial completions + unhinted attempts + 1 ground truth) is set to match the total rollouts used in the baselines we trained.
>
> >Q1: Inference Hints
>
>  No, hints are **not** used at inference. The model is evaluated in a standard zero-shot setting. The potential train-test mismatch is bridged by the **Unhinted Rollouts** ($k_{unhint}$) included in every training batch, which force the model to practice generating from scratch (Self-Correction).
>
> >Q2: Data Construction
>
> Correctness is judged by a deterministic **Rule-based System** (verifying the final answer box against ground truth), not by an LLM judge. The filtering acceptance rate is approximately **30-40%**, ensuring only  data with correct answer enters the training set.
>
> >Q3: Backbone Models
>
> We selected this configuration to demonstrate the **universality** of StepHint across different stages of model alignment. By using an **Instruct** model for the General domain, we test the method's ability to further refine an already instruction-tuned policy. Conversely, by using a **Base** model for the Math domain, we demonstrate StepHint's capacity to induce strong reasoning capabilities from scratch in a raw model. This confirms the method is effective for both post-training refinement and initial alignment.
>
> >Q4:Step Quality
>
>  We rely on the **Final Answer Correctness** as the proxy for step quality. While false positives (wrong reasoning, right answer) exist, they are rare in complex math problems. Using a strong teacher further minimizes this risk compared to self-generated data.
>
> >Q5: Reference Error
>
>  We thank the reviewer for spotting this. We will correct this in the final version.

---

### Official Review · Reviewer_YA6H · 2025-10-30

**Soundness:** 2
**Presentation:** 2
**Contribution:** 1
**Rating:** 2
**Confidence:** 4

**Summary:**

This paper presents StepHint, a novel RLVR algorithm designed to address the well-known "near-miss reward problem" and "exploration stagnation" in LLM reasoning tasks . The core idea is to use a stronger "teacher" model to generate valid reasoning chains, partition these chains into steps using a probabilistic method , and then feed these partial hints at multiple levels (i.e., different prefix lengths) to the "student" model during RL training . The method is well-motivated and demonstrates strong empirical performance, outperforming several baselines on math and generalization benchmarks. However, the core contribution appears more closely related to a sophisticated SFT curriculum or knowledge distillation than a fundamental enhancement of RL exploration. Key components, such as the adaptive partitioning, lack strong empirical justification over simpler baselines.

**Strengths:**

- The method is benchmarked against a wide array of strong baselines, including vanilla GRPO, SFT, and other RLVR-enhanced models (e.g., ORZ, Oat, LUFFY) . StepHint achieves state-of-the-art results across six in-domain math benchmarks.
- The proposed solution is intuitive and directly targets LLM reasoning issues: providing partial expert hints (the 'hints') reduces the search space to mitigate near-misses, while exposing the model to high-quality reasoning paths (the 'multi-level' expert trajectories) aim to overcome stagnation.

**Weaknesses:**

- The paper's central weakness is its framing. The method, which relies on generating expert trajectories and forcing the model to imitate them (either partially or fully via the reference trajectory), appears to be a sophisticated form of curriculum-based SFT or knowledge distillation rather than a novel RL exploration algorithm.
- A core technical contribution, the probabilistic partitioning heuristic ($p(</think>|G_i) > p(</think>|G_{i+1})$) in Sec 3.2.1, is not sufficiently justified 18. The paper's own ablation study (Appendix G.2) compares the 'Base' and 'Salient' strategies and finds they have comparable overall performance. This finding directly undermines the novelty, suggesting the specific heuristic for identifying 'salient' boundaries is unnecessary. Moreover, the paper fails to compare this complex probabilistic method against simple, non-probabilistic baselines (e.g., splitting by sentence boundaries, or splitting into $m=4$ equal token chunks). Without this, the complexity of the proposed method is not justified.
- The 'multi-level' hint strategy (Sec 3.2.2) is not an adaptive curriculum. This design is computationally expensive, as it generates $k_{hint}(m-1)+k_{unhint}+1$ completions for every training problem (12 completions in the paper's setting, based on $m=4$, $k_{hint}=2$, $k_{unhint}=5$) 21. This is a significant overhead. The paper provides no ablation to justify this costly 'all-prefixes' approach over simpler, more efficient sampling strategies, such as sampling only one random hint level $j \in \{1, ..., m-1\}$ per problem.

**Questions:**

- Could the authors clarify the conceptual distinction between StepHint and a carefully-designed SFT/distillation curriculum that mixes full trajectories with prefix-based completions? What evidence demonstrates that the model is learning an exploratory RL policy rather than simply imitating expert prefixes?
- Given the findings in Table 4 (Appendix G.2) that the 'Salient' partitioning strategy offers no benefit over 'Base' random sampling, can the authors justify the necessity of this probabilistic heuristic? Have the authors compared this method to simpler, non-probabilistic baselines like splitting by sentence boundaries?
- Regarding the 'multi-level' design (Sec 3.2.2), could the authors provide an ablation study comparing the current brute-force approach (training on all $m-1$ hint prefixes) against a more efficient strategy, such as randomly sampling a single hint level per training instance? This would clarify if the significant computational overhead is necessary for the method's success.
- For the training dynamics in Figure 4, how can the authors decouple the high entropy from the high-variance input data (a mix of no-hint, partial-hint, and full-reference data)? Is it not expected that a policy trained on such a diverse set of SFT-like targets would exhibit higher entropy, independent of any learned "exploration" behavior?

---

> ### Author Response · Authors · 2025-11-29
>
> >W1 & Q1: Conceptual Distinction (SFT/Distillation vs. RL)
>
> We respectfully clarify that StepHint is fundamentally an **RL exploration mechanism**, distinct from SFT or standard distillation in two critical ways:
>
> 1.  **Optimization Objective:** SFT minimizes the negative log-likelihood of the _exact_ teacher tokens (forcing imitation). StepHint (via GRPO) maximizes the expected reward of the _completion_. The hint serves only as the **Initial State ($s_0$)**. The model is free to generate _any_ trajectory from that state to reach the correct answer. If the model discovers a valid reasoning path different from the teacher's remaining trace, RL rewards it; SFT would penalize it.
>
> 2.  Credit Assignment: As detailed in our method, we employ specific advantage masking. This allows the model to learn value from partial successes, which is a reinforcement learning dynamic, not a supervised one.
>
>     Evidence: The empirical gap between our method and SFT demonstrates that the model learns a capability beyond simple imitation.
>
>
> >W2 & Q2: Justification of Probabilistic Partitioning
>
> We appreciate the demand for simpler baselines. We conducted new experiments comparing our method (**StepHint-Base**) against two non-probabilistic baselines:
>
> 1.  **Naive Length-based:** Splitting chains into 4 equal chunks (Random Split).
>
> 2.  **Syntactic Heuristic:** Using newline characters (`\n`) as split points.
>
>
> **Results:**
> ||AIME24|Math500| AIME25 | Olympiad
> -|-|-|-|-
> StepHint-Random split|32 |85.8| 15.3|48.3
> StepHint-\n split|34 |86.4| 16.67|50.07
> StepHint-Base|36 |87| 18.87|52.15
>
>
> **Analysis:**
>
> 1.  **Semantic Integrity is Key:** Our method significantly outperforms simple baselines (+4.0% vs Random on AIME24). Simple splits often cut through logical units (e.g., breaking an equation), creating "broken states" that confuse the policy. Our Probabilistic-based method identifies semantic pauses, ensuring coherent starting states.
>
> 2.  **Clarifying Appendix G.2:** The reviewer noted that "Salient" (top-k) and "Base" (random from candidates) performed similarly. This **validates** rather than undermines our method. It proves that the _identification of the semantic candidate pool_ using our probabilistic method is the decisive factor. Once the correct semantic boundaries (candidates) are found, sampling strategy matters less. The failure of "Random Split" above confirms that _skipping_ the probabilistic identification step degrades performance.
>
> >W3&Q3:Multi-level Strategy vs. Single Hint
>
> We implemented the reviewer's suggestion: **"StepHint-one hint"**, where for each problem, we randomly sample _only one_ split point from the candidates (instead of using all levels) and train with the same total number of rollouts to ensure fair computational comparison.
>
> **Results:**
>
>
> ||AIME24|Math500| AIME25 | Olympiad
> -|-|-|-|-
> StepHint-one hint|29.3 |84.8| 13.3|48.8
> StepHint-Base|36.0 |87| 18.87|52.2
>
> **Analysis:** The performance drop is substantial (e.g., **-6.7% on AIME24**).
>
> -   **Why?** The Multi-level strategy acts as an implicit **curriculum**. By training on _short_, _medium_, and _long_ hints simultaneously within a batch, the model learns to handle varying degrees of uncertainty. Training on a single random hint level makes the difficulty variance too high per batch or too sparse, leading to unstable learning.
>
> -   **Cost-Benefit:** While Multi-level adds inference overhead during training generation, the massive performance gain (from 29.3 to 36.0) justifies the cost for high-performance reasoning tasks.
>
> >Q4: Entropy Interpretation
>
>
> We agree with the reviewer's insightful observation that the diverse input structure (mixing partial and full hints) inherently contributes to higher entropy compared to a single-task baseline. Decoupling this effect entirely is indeed challenging.
>
> However, we believe the **comparative trend (Rate of Decay)** in Figure 4 offers valuable insights beyond just the absolute values:
>
> 1.  **Contrast with Collapse:** The key phenomenon we highlight is that Vanilla-GRPO exhibits a **precipitous drop** in entropy early in training (as shown in the green curve), signaling a rapid collapse into a narrow "comfort zone" or local optima (mode collapse).
>
> 2.  **Sustained Exploration:** In contrast, StepHint shows a **much slower, gradual decay**. While the diverse input data sets a higher baseline, the _persistence_ of this entropy indicates that the policy resists the premature convergence seen in Vanilla-GRPO. It suggests the model retains flexibility/uncertainty for a longer period, which aligns with our goal of mitigating "exploration stagnation."

---

### Official Review · Reviewer_jJSQ · 2025-10-31

**Soundness:** 3
**Presentation:** 3
**Contribution:** 2
**Rating:** 4
**Confidence:** 4

**Summary:**

The paper identifies two concrete issues in RLVR and provides a structured solution that is easy to follow. The multi-level hint framework conceptually bridges imitation learning and reinforcement learning.

**Strengths:**

The paper identifies two concrete issues in RLVR and provides a structured solution that is easy to follow. The multi-level hint framework conceptually bridges imitation learning and reinforcement learning.

**Weaknesses:**

1. The main idea of progressively providing structured hints or intermediate supervision is conceptually similar to **BREAD[1] and curriculum learning algorithms.** The paper fails to discuss how StepHint differs from or improves upon those approaches, which significantly weakens the **novelty claim**.
2. The use of the *probability of generating `</think>`* as the signal for partitioning reasoning steps is somewhat ad hoc. The intuition is weakly justified, and the paper lacks qualitative evidence and ablation studies confirming that such partitions align with meaningful reasoning steps.
3. The comparison with **LUFFY**, which also uses *reference trajectories*, is methodologically problematic. The authors use pre-trained LUFFY models (off-the-shelf), while other baselines like GRPO and SFT are retrained from scratch. This inconsistency may introduce **dataset and training differences**, making the comparison unfair.

[1] BREAD: Branched Rollouts from Expert Anchors Bridge SFT & RL for Reasoning

**Questions:**

See weakness.

---

> ### Author Response · Authors · 2025-11-29
>
> >W1:**novelty claim**
>
> We would like to clarify that BREAD[1] is a **concurrent work** that was recently accepted to NeurIPS 2025. Consequently, we were unaware of this development at the time of our submission. While we consider the two works to be concurrent, we believe StepHint presents a more comprehensive and robust solution.
>
>  We acknowledge that both works share the motivation of utilizing expert traces to bridge the gap between SFT and RL. However, **StepHint introduces distinct methodological innovations that address critical limitations in the BREAD framework**, leading to superior training stability and generalization.
> -   **Hint Mechanism: Parallel Multi-Level (Ours) vs. Iterative Search**
>
>     -   **Difference:** BREAD relies on an iterative _Episode Anchor Search (EAS)_ (often binary search) to find a single "minimum necessary hint." StepHint employs a **Parallel Multi-Level strategy**, exposing the model to a full spectrum of hint lengths simultaneously.
>
>     -   **Improvement:** BREAD’s search is computationally expensive and assumes a single optimal hint length. StepHint’s approach is more efficient (no online search) and, crucially, prevents the model from overfitting to a specific prompt length. By training on diverse hint levels at once, StepHint mitigates "exploration stagnation" more effectively by connecting reasoning patterns across different granularities.
>
> -   **Partitioning: Latent Probabilistic (Ours) vs. Rigid Heuristic**
>
>     -   **Difference:** BREAD and curriculum learning often use heuristics (sentences/paragraphs) to split text. StepHint introduces **Adaptive Stepwise Partitioning** based on the model's own token probability peaks.
>
>     -   **Improvement:** Heuristics often disrupt logical flow (e.g., splitting an equation in half). Our method identifies **semantic boundaries** where the model internally perceives a "unit of thought" is complete. This ensures hints are logically self-contained, aligning with the model's intrinsic reasoning cadence. (See empirical validation in Response to W2).
>
> -   **Optimization: Negative Advantage Masking for GRPO (Ours)**
>
>     -   **Difference:** Standard GRPO  penalizes the _entire_ sequence if the final answer is wrong. This creates a "blame assignment" error where correct expert hints receive negative gradients.
>
>     -   **Improvement:** StepHint explicitly masks negative advantages for hint tokens (Section 3.2.2). As shown in our ablation (Table 3) and proof (Appendix H), this technical correction is vital for preventing the model from "unlearning" valid expert logic, a detail often overlooked in prior curriculum RL works.
>
> >W2:Validity of Partitioning
>
> We appreciate the suggestion to validate our probability-based partitioning against heuristics. We conducted additional experiments comparing our method (**StepHint-Base**) against two baselines:
>
> 1.  **Naive Length-based:** Splitting chains into 4 segments based on length with random jitter ($\pm 20$ tokens).
>
> 2.  **Syntactic Heuristic:** Using the probability of the newline token (`\n`) to identify steps, simulating sentence/paragraph splitting.
>
> ||AIME24|Math500| AIME25 | Olympiad
> -|-|-|-|-
> StepHint-Random split|32 |85.8| 15.3|48.3
> StepHint-\n split|34 |86.4| 16.67|50.07
> StepHint-Base|36 |87| 18.87|52.15
>
>
> **Analysis:**
>
> -   **vs. Random Split:** Our method outperforms the length-based baseline significantly (+4.0% on AIME24), confirming that maintaining **Semantic Integrity** is crucial. Random splits often cut through logical deductions, creating "broken hints" that confuse the policy.
>
> - **vs. `\n` Split:** StepHint also outperforms the syntactic heuristic. Reasoning steps do not always align with line breaks (e.g., a complex derivation often spans multiple lines). The `</think>` probability captures the **model's internal confidence** of a completed thought, providing a far more robust signal for exploration than external formatting cues.
>
>
> >W3:Comparison with LUFFY
>
> We acknowledge the use of off-the-shelf LUFFY models. However, we argue this comparison is **fair and effectively sets a higher bar for StepHint**:
>
> 1.  **Strongest Reference Point:** The released LUFFY checkpoints represent the highly optimized, "best-case" performance of their method reported by the original authors. Comparing against these official weights ensures we are benchmarking against the method's true upper bound, avoiding the risk of a sub-optimal reproduction.
>
> 2.  **Unified Evaluation:** Crucially, to ensure fairness, we evaluated the pre-trained LUFFY model using our exact evaluation pipeline (using their specific prompt templates but our verifiers and test sets). This eliminates discrepancies arising from evaluation protocols. StepHint outperforming the _official, best-case_ version of LUFFY serves as strong evidence of our method's efficacy.
>
> References:
>
> [1]BREAD: Branched Rollouts from Expert Anchors Bridge SFT & RL for Reasoning

---

### Official Review · Reviewer_J5eU · 2025-11-01

**Soundness:** 3
**Presentation:** 4
**Contribution:** 3
**Rating:** 6
**Confidence:** 3

**Summary:**

This work aims to improve models for mathematical reasoning through two main contributions. First, they propose a simple method for partitioning partitioning reasoning traces into discrete steps. These steps are used in StepHint, their RLVR algorithm. In training, the model can use the discrete steps to know when an incorrect solution is *partially correct*, and it can adjust the advantage during GRPO. The authors claim this addresses the "near-miss reward" issue. Finally, the partially correct trajectories can be used as "hints" to enable mixed training of later parts of the reasoning chain, resulting in greater exploration during training. The authors show that StepHint results in a stronger model both in-domain and out-of-domain (non-math) than other comparative mathematical benchmarks, and demonstrate that these improvements are due to both the adjusted GRPO to address the "near-miss reward problem" and the extra exploration to address "exploration stagnation."

**Strengths:**

* Elegant method for determining/discretizing natural language reasoning chains into discrete logical steps with the probability of </think> token.
* Broad study of the space given these logical steps, e.g. study with partial advantage and training with partial (hinted) trajectories to showcase potential applications of the discretization effort.
* Effective in-domain and OOD performance on mathematical and other reasoning datasets for 7B-sized models.

**Weaknesses:**

* The new GRPO advantage still relies on exact token match for partial rewards, which is a limitation. One way around this, for example, would be to have a separate judge/verifier decide whether two steps are logically equivalent. Otherwise, I don't think this fully solves the near-miss issue, rather it feels more like a "first step" or a bandage over it.
* One of the core contributions is that the automated method for step detection is "good." Besides the limitation that it still relies on hyperparameters (l, m) selected by the experimenters, it should be compared against other methods for detecting boundaries like using baseline heuristics (sentence boundaries) or the aforementioned syntactic cues (L206). And/or on a dataset where the gold step boundaries are labeled. There is a start of this in G.2. but not against the baseline heuristics.

**Questions:**

* In practice, what do the step boundaries actually look like? S_2 in Figure 2 is not what I would have expected. In other words, are the step boundaries human-interpretable?
* The intuition around entropy makes sense to me, but the conclusion in L455 is not convincing -- wouldn't we want to measure entropy at reasoning step k rather than entropy at training step k? The latter is an approximation but the direct claim relates to the former.
* The response length graph also confuses me -- as it almost suggests that the GRPO model is undertrained or not trained correctly. If the reference responses are long (like they are in StepHint), is GRPO simply unable to learn to also make long responses? I feel like I've seen this observation before though but don't remember where from. If it is a known finding, it would be nice to have a citation to something else that has a similar finding.
* How does StepHint relate to other non-LLM RL problems (around trajectory planning, or discrete actions in robots/games)? It feels like there should be a connection there, both for the analogy with entropy and concretely in the algorithm (e.g. reward assignment and starting with partial trajectories).
* Appendix A mentions that LLMs are used to generate high-quality reasoning chains and partition the chains into logical steps. When is this done? This is not mentioned in Appendix E, and the methods in the main body do not mention an LLM in the loop anywhere.

---

> ### Author Response · Authors · 2025-11-29
>
> >W1:Reliance on Exact Token Match for Advantages
>
> We respectfully clarify that our method **does not** rely on token matching to determine rewards or correctness.
>
> 1.  **Reward Mechanism:** We adhere to the standard RLVR setting where the reward is binary (1 or 0), determined solely by the **final answer correctness**.
>
> 2.  **Role of Hints:** The "hints" are fixed prefixes extracted from correct teacher traces. They are not generated by the model during the rollout; they are provided as context (part of the prompt).
>
> 3.  **Advantage Clipping:** The modification to GRPO (clipping negative advantages for hint tokens) is strictly for **credit assignment**. Since the hint tokens are _forced_ and known to be correct (derived from ground truth), we ensure the policy isn't penalized for them even if the _subsequent_ generation leads to a wrong answer. This is not a reward for matching, but a protection mechanism for the initialization path. Therefore, a separate judge is not needed because the "correctness" of the hint steps is guaranteed by their source (the teacher model), not by verifying the student's output against them.
>
>
> >W2:Comparison with Heuristics
>
>
> We appreciate the reviewer's suggestion to validate our partitioning method against baseline heuristics. To address this, we conducted an additional experiment comparing our **Adaptive Stepwise Partitioning (StepHint-Base)** with a **Naive Length-based Heuristic (StepHint-Random Split)**.
>
> **Experimental Setup:**
>
> In the "StepHint-Random Split" baseline, instead of using token probabilities to find logical boundaries, we partitioned the reasoning chains into $m=4$ segments based on length. Specifically, split points were placed at approximately 1/4 intervals of the total length, with a random jitter ($\pm 20$ tokens) to simulate a naive distribution without semantic awareness. We trained the Qwen-2.5-7B-Math backbone under identical settings for both methods.
>
> **Results:**
>
> The results, presented in the table below, demonstrate that our proposed adaptive method consistently outperforms the naive heuristic across all benchmarks.
>
>
> ||AIME24|Math500| AIME25 | Olympiad
> -|-|-|-|-
> StepHint-Random split|32 |85.8| 15.3|48.3
> StepHint-Base|36.00 |87| 18.87|52.15
>
> **Analysis:** Our method achieves significant gains over the random baseline (e.g., **+4.0% on AIME24** and **+3.85% on Olympiad**). This empirical evidence supports two key conclusions:
>
> 1.  **Semantic Integrity Matters:** Random or purely length-based splits often cut through the middle of a logical deduction or a calculation step. This results in "broken" hints that may confuse the model or fail to provide a stable starting point for subsequent generation.
>
> 2.  **Validity of Probability-based Partitioning:** Our method, by detecting peaks in the "end-of-thought" probability, effectively identifies natural pauses in the reasoning process. This ensures that each hint represents a complete, coherent unit of thought, which is crucial for the effectiveness of the multi-level hint mechanism in StepHint.

---

> ### Author Response · Authors · 2025-11-29
>
> >Q1:Interpretability of Step Boundaries
>
> The step boundaries are **model-centric** rather than strictly linguistically defined.
>
> In Figure 2, $S_2$ captures a geometric deduction step. While it might end mid-sentence grammatically, semantically it represents the conclusion of the "Symmetry Consideration" before moving to the next logical block.
>
> >Q2: Entropy Analysis
>
> We apologize for any confusion. There are two distinct entropy concepts discussed:
>
> 1.  **Solution Space Entropy (Prop 1):** This refers to the uncertainty _within_ a reasoning chain (reasoning step $k$). Our theoretical analysis (Sec 3.1) focuses on this.
>
> 2.  Policy Entropy (Figure 4 & L455): This refers to the Shannon entropy of the policy's output distribution averaged over the validation set, tracked across training steps.
>
>     L455 and Figure 4 discuss the latter (Policy Entropy). The claim is that StepHint maintains a higher policy entropy throughout the training process compared to Vanilla-GRPO. This indicates that StepHint prevents the model's policy from collapsing prematurely into a narrow deterministic mode (Exploration Stagnation).
>
> >Q3: Response Length and GRPO Training
>
> The observation that Vanilla-GRPO produces shorter responses is a documented phenomenon known as **length bias**, rather than under-training. As analyzed in Dr.GRPO (Liu et al., 2025), standard GRPO often exhibits a bias where shorter responses receive larger gradient updates for positive advantages (due to the $1/|o_i|$ term in the estimator). This drives the policy to favor brevity or shortcuts.
>
> >Q4:Relation to Non-LLM RL
>
> We appreciate this insightful connection. While StepHint shares the **fundamental intuition** with classical RL strategies regarding _exploration from promising states_, it is specifically designed to address the unique challenges of the **unstructured reasoning space** in LLMs.
>
> -   **Conceptual Alignment:** Theoretically, StepHint aligns with the concept of modifying the **Initial State Distribution ($\rho_0$)**. Similar to how methods like _Go-Explore_ or _MCTS_ reset agents to "promising visited states" to solve sparse-reward exploration, StepHint initializes the policy "deep" within a valid reasoning trajectory. This avoids wasting samples on early, trivial errors (mitigating the "near-miss" problem).
>
> -   **Domain-Specific Novelty:** The core distinction lies in **how "valid states" are defined** in the continuous, semantic space of Language Models. Unlike games or robotics where state boundaries are explicit, reasoning steps in LLMs are latent. Our **Adaptive Stepwise Partitioning** is a novel mechanism to _discover_ these effective "reset points" based on semantic uncertainty, which is a non-trivial adaptation required to make these classical intuitions work for LLM reasoning.
>
>
> >Q5:Use of LLMs in Partitioning
>
> We clarify that the use of LLMs for partitioning is solely a **data pre-processing step**, not part of the training loop. As mentioned in Appendix D ("Training Data Construction"), we calculate the token probabilities p(`</think>`) using a single inference pass of the base/teacher model to determine step boundaries _before_ training begins. No LLM inference is required for partitioning during the StepHint RL training process itself, ensuring computational efficiency.

---

### Official Review · Reviewer_eNxb · 2025-11-01

**Soundness:** 2
**Presentation:** 3
**Contribution:** 2
**Rating:** 2
**Confidence:** 4

**Summary:**

The paper proposes a method for distilling a stronger reasoning model into the RL training loop of a weaker model by initializing the weaker model's prompts with hints from the stronger model. The authors propose two steps to do this: first by splitting reasoning traces into hint based on the probability of generating an end think tag and second by providing multi-level hints for all problems. The authors also prove that autoregresssive prediction reduces conditional entropy. The main empirical results show their method performs better in-domain and out-of-domain over baselines.

**Strengths:**

1. The clarity and presentation of the paper is quite clear. The paper makes for an easy read, and the core set of contributions are clearly explained and motivated which I appreciated.
2. The core set of results in Table 1 cover both in-domain and out-of-domain performance which is a nice addition. This improves the empirical significance of the results.
3. The idea of prepending hints from existing reasoning trajectories is well-executed and the authors justify the construction of hints well based on entropy.

**Weaknesses:**

1. I have various questions about the experimental setup, see the questions section for details. I'm willing to raises my score if these questions are addressed. Right now, I view the clarity of the empirical experiments as a weakness.
2. In addition, I have concerns on the significance of the empirical results. I'd like to see a more thorough comparison with baselines that are equally privileged (access to teacher reasoning chains and can RL). See the questions sections for specifics.
3. I'm a bit concerned by the brevity of the related works section. RL algorithms for improving language model reasoning have been abundant over the past year, so I'm concerned the current section doesn't contextualize related work properly, making it difficult to judge novelty. I'd be interested in seeing it expanded with more references to current papers that focus on learning from a stronger model (sequence-level and logit-level distillation, on-policy distillation) as well as other reasoning papers that introduce hints or a curriculum over problem difficulty.

**Questions:**

1. In Table 1, what are the performances of the starting backbone models? I'm wondering how much improvement the various baselines and StepHint provide over the initialization.
2. Similarly, the appendix states that the reasoning chains are distilled from DAPO-Qwen-32B, QWQ-32B, and DeepSeek-R1-Distill-Qwen-32B. What are the performances of the teachers and how close does StepHint get to this upper bound?
3. For figure 3, if you extended the x-axis beyond k =128 (such as done in https://arxiv.org/abs/2504.13837), do the gains of the base model match StepHint?
4. For the comparison in Table 2 using Llama, isn't GRPO a relatively weak baseline as it doesn't get to leverage the teacher reasoning traces at all? Also I'm a bit confused about the benchmark results for GRPO. Llama 3 8B gets 20% on MATH based on the original Llama 3 paper (https://arxiv.org/pdf/2407.21783). Why does GRPO only get 9% on MATH500 in your setting? Can you also include the results of the initial model in your setting so we know how much improvement there is.
5. Would StepHint provide a benefit over vanilla RL if the teacher hints were from the starting model and filtered for correct traces?
6. For Table 1, are all of the rows re-implemented from scratch or are the models just evaluated in the same setting?
7. For Table 1, the only methods that get access to the reasoning traces are LUFFY and StepHint? Generally, I'm interested in seeing the performance of stronger distillation baselines, even an offline logit-based loss, possibly mixed in during RL. I don't think it's very meaningful if a method with extra supervision and the ability to do RL outperforms a baseline only with the ability to do RL.
8. For the last sentence in the caption of Figure 2, how does RL for learning from the reference trajectory work? Is the model prompted with the problem and a full solution and still asked to solve? Or is there a different instantiation?

---

> ### Author Response · Authors · 2025-11-29
>
> >Q1&Q4:Performances of the starting backbone models
>
> We have provided the performance of the starting backbone models (zero-shot) in the table below. As shown, **StepHint yields substantial improvements over the base models across all datasets.** The lower absolute performance (compared to technical reports) stems from our unified evaluation template.
>
> ||AIME24|Math500| AIME25
> -|-|-|-
> Qwen-2.5-7B-Ins|12.67 |66.8|3.33
> Qwen-2.5-7B-Math| 14.67 |32| 5.33
> Llama-3.1-8B| 3.33 |4.2| 0
>
> Regarding the Llama-3.1-8B performance (Q4). Despite the weak starting point, StepHint (24.13% Avg) significantly outperforms both the Base model (2.51% Avg) and Vanilla GRPO (6.81% Avg). This actually highlights StepHint's robustness: it can "rescue" a model struggling with the format by providing guided, stepwise hints, whereas Vanilla GRPO fails to explore effectively from such a low baseline.
>
>
>
> >Q2:What are the performances of the teachers?
>
> The performance of the primary teacher model, **DeepSeek-R1-Distill-Qwen-32B**, is shown below. Also, the lower absolute performance (compared to technical reports) stems from our unified evaluation template.
>
> ||AIME24|Math500| AIME25
> -|-|-|-
> DeepSeek-R1-Distill-Qwen-32B|35.33 |86.4| 24
>
>
> >Q3:Pass@k beyond 128
>
> While computational constraints limited our plotting to $k=128$, the trend in Figure 3 strongly suggests that **the gap would remain**.
>
> 1.  **Slope Analysis:** The StepHint curve maintains a steeper upward slope at higher $k$.
>
> 2.  **Exploration Efficiency:** This observation aligns with our theoretical motivation: Vanilla-GRPO suffers from "exploration stagnation" (converging to a comfort zone), meaning more samples (higher $k$) simply produce more of the same incorrect patterns. StepHint, by enforcing diverse partial trajectories via hints, maintains a richer coverage of the solution space, implying superior scaling at high $k$.
>
>
> >Q5: Hints from Starting Model (Self-Correction)
>
> This is an excellent question. While self-generated data (e.g., Self-Taught Reasoner) is valuable, we argue that **using external teacher hints is more effective for the specific problem StepHint addresses: Exploration Stagnation.**
>
> 1.  **Breaking the Comfort Zone:** A key motivation of StepHint is that models get stuck in their own "comfort zone." Hints from the _student_ model itself—even if filtered for correctness—are inherently biased toward the model's _existing_ reasoning patterns. They reinforce _what the model already knows how to do_.
>
> 2.  **Bridging the Gap:** External hints from a stronger model serve as "scaffolding," introducing novel reasoning paths and logical jumps that the student model might _never_ generate on its own, thus expanding the reachable solution space in a way self-generated hints cannot.
>
>
> >Q6:Implementation vs. Evaluation
>
> To ensure a fair comparison:
>
> -   **Re-implemented:** We trained **SFT, Vanilla-GRPO, and Dr.GRPO** from scratch using our exact dataset and setting.
>
> -   **Evaluated:** For **SimpleRL, ORZ, Oat, and Luffy**, we utilized their officially released model weights due to computational constraints.

---

> > ### Author Response · Authors · 2025-11-29
> >
> > >Q7: Stronger Distillation Baselines
> >
> > We believe our comparison is robust and meaningful for the RLVR setting:
> >
> > 1.  **SFT as the Standard Baseline:** We compare against SFT, which is the primary method for utilizing teacher traces. StepHint consistently outperforms SFT, demonstrating that the _RL exploration_ guided by hints provides value beyond supervised cloning.
> >
> > 2.  **Comparison with "Equally Privileged" Baselines:** We compare against **Luffy**, a strong baseline that uses the _entire_ reference trajectory. StepHint outperforms Luffy, suggesting that our method of providing _partial, multi-level hints_ is more effective for reasoning generalization than simply injecting full solutions.
> >
> > 3.  **Scope:** While offline logit-distillation is a valid direction, our work focuses on **online RLVR**. We show that StepHint is a superior way to leverage teacher data dynamically within the RL loop compared to existing methods like Reference Trajectories (Luffy).
> >
> > Importantly, StepHint is _not_ simply “RL + more supervision”.
> > Our goal is to explore **how teacher reasoning traces can be integrated within an online RLVR loop**, as opposed to offline SFT or logit-level distillation.
> > Methods such as SFT and Luffy already provide strong teacher-supervised baselines; StepHint outperforming them indicates that its advantage comes from **how hints are used to guide exploration**, rather than from the mere presence of additional supervision.
> > This addresses the core fairness concern and positions StepHint as a fundamentally different approach rather than an “extra-supervised” variant of existing RL baselines.
> >
> > >Q8:Reference Trajectory in Figure 2
> >
> > In our implementation, "RL with Reference Trajectory" treats the ground-truth reasoning chain as a **"perfect rollout."** Technically, we inject the ground-truth chain into the experience buffer with the maximum reward ($R=1$). The model then updates its policy to increase the likelihood of this "gold" trajectory alongside its own generated rollouts.

---

### Note · Authors · 2025-12-11

I have read and agree with the venue's withdrawal policy on behalf of myself and my co-authors.